# Cell-Secreted Vesicles: Novel Opportunities in Cancer Diagnosis, Monitoring and Treatment

**DOI:** 10.3390/diagnostics11061118

**Published:** 2021-06-19

**Authors:** Cristina Catoni, Veronica Di Paolo, Elisabetta Rossi, Luigi Quintieri, Rita Zamarchi

**Affiliations:** 1Veneto Institute of Oncology IOV-IRCCS, Padua, Italy; cristina.catoni@iov.veneto.it (C.C.); rita.zamarchi@unipd.it (R.Z.); 2Laboratory of Drug Metabolism, Department of Pharmaceutical and Pharmacological Sciences, University of Padua, Padua, Italy; veronica.dipaolo@unipd.it; 3Department of Surgery, Oncology and Gastroenterology, University of Padua, Padua, Italy

**Keywords:** extracellular vesicles, cancer biology, liquid biopsy, therapeutic tools

## Abstract

Extracellular vesicles (EVs) are important mediators of intercellular communication playing a pivotal role in the regulation of physiological and pathological processes, including cancer. In particular, there is significant evidence suggesting that tumor-derived EVs exert an immunosuppressive activity during cancer progression, as well as stimulate tumor cell migration, angiogenesis, invasion and metastasis. The use of EVs as a liquid biopsy is currently a fast-growing area of research in medicine, with the potential to provide a step-change in the diagnosis and treatment of cancer, allowing the prediction of both therapy response and prognosis. EVs could be useful not only as biomarkers but also as drug delivery systems, and may represent a target for anticancer therapy. In this review, we attempted to summarize the current knowledge about the techniques used for the isolation of EVs and their roles in cancer biology, as liquid biopsy biomarkers and as therapeutic tools and targets.

## 1. Introduction

For a long time since their discovery, the role of extracellular vesicles (EVs) in cancer remained poorly understood. EVs are currently considered the main transporters of specific cargoes, including the molecular components of parent cells, thus mediating a wide variety of cellular activities in both normal and neoplastic tissues. These vesicles are secreted by several cell types (e.g., tumor cells, macrophages and fibroblasts) and are widely distributed in the blood, urine, ascites, synovial fluid, breast milk and other bodily fluids [1]. They have been identified as key messengers of intracellular communication in healthy and neoplastic cells. Tumor cell-derived EVs have become a popular research topic in the field of cancer studies [2], and there have been many reports on EVs in cancer.

Liquid biopsy monitors tumor development through non-invasive sampling. Recently, EVs have started to attract attention as a component of liquid biopsy and among disease biomarkers. This is because EVs have multiple advantages, for example, their abundance in biofluids and protection of proteins and nucleic acids from degradation through their lipid bilayer membrane [3]. They have emerged as among the most promising liquid biopsies, and several studies have demonstrated that EVs could reflect tumor development and progression [4].

At the same time, EVs are very interesting for theragnostic purposes. In fact, their inhibition, control of EV-related gene expression and hemofiltration of EVs all prevent or reduce intercellular communication between cancer cells [5]. Another way to use EVs could be as drug delivery nanocarriers, cancer vaccines, cell surface modulators, therapeutic agents and therapeutic targets [6].

In this review, we sought to summarize the most recent literature on EVs regarding isolation techniques as well as their roles in cancer biology, diagnosis and therapy.

## 2. Extracellular Vesicles: Classification and Characteristics

The term EV encompasses a highly heterogeneous group of membrane-delimited nanoparticles, which are secreted by all cell types in both eukaryotic and prokaryotic organisms [7]. The International Society for Extracellular Vesicles (ISEV) has attempted to summarize the “minimal experimental requirements for definition of extracellular vesicles and their functions” (MISEV) [8,9]. ISEV suggests using the term “extracellular vesicles” as the “generic term for particles naturally released from the cell that are delimited by a lipid bilayer and cannot replicate” and to enrich the terminology “EV” by specifying physical characteristics (e.g., size and density), their biochemical composition or cell of origin (Table 1) [8]. However, some authors consider the formulated guidelines to be vague, and that currently the nomenclature remains unclear [10].

When referring to EV-mediated intracellular communication in the regulation of normal physiological processes, or in the pathological processes of many diseases including cancer [11,12], several terms are found in the literature (e.g., exosomes, ectosomes, microparticles, microvesicles, membrane particles, separated microvesicles, exosome-like particles, apoptotic vesicles, prominosomes, prostasomes, migrasomes and oncosomes) [13,14]. All these terms demonstrate a lack of a unique criterion for EV classification.

A worldwide survey conducted by ISEV in 2019 found that most of EVs are harvested from blood plasma, serum, urine and cerebrospinal fluid [1]. Much less frequently, EVs were purified from bronchoalveolar lavage, peritoneal fluid or semen [1]. The results of this survey are in agreement with those of the first worldwide survey published by Gardiner et al. in 2016 [15].

Blood plasma is considered the preferred source of EVs, since serum contains additional vesicles which are released during clot formation in the course of its isolation. However, it should be taken into account that there is currently no method capable of isolating EVs only, and in particular, protein aggregates, lipoproteins and platelets represent the most common contaminants [16].

Most authors distinguish exosomes, microvesicles (MVs) and apoptotic bodies as the major types of EVs based on their cellular origin, as well as their physiochemical and biomolecular properties [13,17], although no straightforward criteria exist to classify, isolate and identify the subpopulations of cell-derived vesicles. In addition, other types of EV populations have been identified with biochemical and compositional characteristics typical of certain pathological conditions (e.g., large oncosomes in cancer) [16,18]. Exosomes are generated from late endosomes called multivesicular bodies (MVBs) and range in size from 30 to 120 nm. MVs known as “shedding microvesicles” or “ectosomes” are formed by budding from the plasma membrane. Accordingly, MVs vary in size with a minimum diameter of 100 nm but lack an upper size limit which could reach several microns [19]. MV membranes have higher levels of cholesterol, diacylglycerol and phosphatidylserine than exosomes. Apoptotic bodies are larger in size (500–4000 nm) and represent the main type of vesicles released during cell apoptosis. Similar to MV, apoptotic bodies exhibit phosphatidylserine on their surface, however, unlike all other EVs, these carry fragmented genomic DNA and cell organelles [13,20].

Although the challenge for many researchers is identifying the specific markers for different EV subpopulations, a unique association between molecular markers and subtypes of EV remains to be established. Kowal et al. proposed a categorization of EVs which could be applied to EVs from both cell culture medium and biological fluids obtained after differential centrifugations followed by either flotation or immunoisolation with three tetraspanins-antibodies (i.e., CD9, CD63 and CD81) [21]. This approach allowed the classification of EVs as (I) large EVs pelleting at low centrifugation speed (2000× *g*); (II) medium EVs pelleting at intermediate speed (10,000× *g*); and (III) small EVs (sEVs) pelleting at high speed (100,000× *g*). Moreover, the sEVs in turn were subdivided in: (IIIa) sEVs coenriched in CD63, CD9 and CD81 and in endosomal markers (i.e., bona fide exosomes); (IIIb) sEVs without CD63 and CD81, but enriched in CD9; (IIIc) sEVs without CD63, CD81 and CD9; and (IIId) sEVs enriched in extracellular proteins or serum proteins.

CD9, CD63 and CD81 are considered good markers for identifying exosomes, however, several papers report the presence of these transpanins not only in exosomes but also in other microvesicles, complicating the discrimination among a different subpopulation of EVs. A possible explanation for this evidence could be the abundant presence of these proteins on the cell surface and thus their incorporation in vesicles which are produced by budding from the plasma membrane [18]. ICAM-1, the stress inducible non-classical major histocompatibility complex (MHC) class-I chain-related antigens A and B (MICA/B), the ligands of major cytotoxic receptors NKG2D, RAET1/ULBP1-5, TRAIL, Fas-L, and PD-L1 are also considered EV markers [22]. More recent studies identified ACTB, MSN and RAP1B as EV pan-markers [23], and annexin A1 as a specific marker for MVs that are directly shed from the plasma membrane [24]. Guan and coworkers recently set up a tandem extraction strategy to obtain metabolites and proteins from the same batch of EVs simultaneously, which enabled a multi-omics differential analysis of exosomes and microvesicles from human plasma. The results of proteomics analysis indicated that the protein compositions between microvesicles and exosomes were certainly different. Indeed, compared to exosomes, 92 proteins including TSPAN32, MAGT1, SNX9 and ABCC1, were upregulated in microvesicles. On the other hand, 20 proteins, including IGHD, CFP and AMBP, were upregulated in exosomes [25].

## 3. EVs: Separation Methods

Over the years, the biochemical and biophysical properties of EVs (i.e., size, mass density, charge and antigen exposure) have been used to develop methods to separate EVs from bodily fluids and the supernatants of cell cultures. In accordance with MISEV2018 [8], here we use the term *separation* because, to date, an absolute purification or complete isolation of the EVs is not feasible. There is not a gold standard protocol to separate EVs since the downstream analyses and the volume of the sample influence the selection of the method.

To date, differential centrifugation remains the most used technique for the separation of EVs, as also emerged in the ISEV global survey conducted at the end of 2019 [1]. Differential centrifugation allows the separation of the EVs according to their size and density by progressively increasing the centrifugal force to pellet in the order of (i) cells and cellular debris; (ii) large EVs; and (iii) small EVs. Numerous protocols are available in the open literature for the separation of EVs; they differ not only in the number of stages but also in the conditions of differential centrifugation (i.e., centrifugation time and/or centrifugal force). In any case, many researchers use as a starting point the Raposo’s protocol [26], which involves a series of sequential centrifugations with an increasing centrifugal force followed by flotation on sucrose density gradients, with the aim of separating exosomes from the conditioned culture media of transformed human B cell lines.

The efficiency of the separation of EVs by differential centrifugation depends both on variables related to centrifugation (i.e., acceleration and characteristics of the rotor) and on the characteristics of the sample (e.g., viscosity). It is known that the high viscosity of the sample reduces the sedimentation efficiency of EVs; therefore, the separation of EVs from plasma or serum requires ultracentrifugation at higher speeds and for longer times than the separation of EVs from cell cultures [13,27]. As it is known that differential centrifugation co-isolate non-EV aggregates of proteins, the density gradient flotation is often used to increase the efficiency of particle separation based on the size, shape and density of the EVs. The most used reagent for density gradient flotation is sucrose, but iodixanol is preferred for the separation of EVs from saliva [28]. Although density gradient centrifugation allows the isolation of EVs of higher purity, a contamination with lipoproteins of a density comparable to that of EVs (i.e., HDL and LDL) has been observed [29,30]. Ultracentrifugation-based methods are unfortunately time-consuming, which limits their clinical use [16].

According to the ISEV global survey in 2019, size-exclusion chromatography (SEC), also known as gel filtration, represents the second most used method for the separation of EVs from biological matrices [1]. In SEC, a porous stationary phase is used to classify particulate matters and macromolecules according to their dimensions. Sample components smaller than the pore size are able to pass throughout the pores, thus resulting in late elution, while components with large hydrodynamic radii (including EVs with a diameter larger than the size cut off), are eluted first. The performance of the SEC is influenced by various parameters including column length, sample volume, and the quality of the column stacking [13]. Compared to ultracentrifugation, SEC produces less mechanical stress on the sample, and preserves vesicle structure and bioactivity [22]. Moreover, SEC is a rapid and relatively inexpensive EVs separation approach, which makes it clinically applicable [31].

In addition to the methods described above, a separation of EVs can be obtained, based on their size, by ultrafiltration, used alone or in association with other separation techniques [32,33]. Ultrafiltration allows the separation of EVs using membranes with pores of different sizes based on the characteristics of the EVs. The filtration of EVs takes place by either applying pressure or by placing the filter in an ultracentrifuge. This method is especially useful for large volume samples having a composition less complex than plasma (e.g., culture media), as it is much faster than differential centrifugation. Notably, ultrafiltration allows to concentrate 100 mL of a sample in approximately 20 min, compared to the 3–9 h required with differential centrifugation [33]. The filtration methods are rapid and highly efficient; unfortunately, there is not yet a reference protocol and the performance could decrease due to the jamming of the filters because of vesicles’ trapping.

The methods described to date for the separation of the EVs are based on physicochemical properties such as size and density. Alternative approaches are represented by immunoaffinity techniques that allow the separation of some subpopulations of vesicles depending on the expression of some surface markers. Most immunoaffinity-based EVs isolation approaches use monoclonal antibodies immobilized on the surface of paper-based devices, beads or chips, which recognize specific proteins of EVs [34,35]. Common targets of immunoaffinity-based capture are tetraspanins (especially CD9, CD63 and CD81), accepted by MISEV2018 as a marker of EV, as well as new markers, and precipitation with hydrophilic polymers such as polyethylene glycol (PEG) [36]. Most immunoaffinity-based EVs isolation approaches use monoclonal antibodies immobilized on the surface of paper-based devices, beads or chips which recognize specific proteins of EVs [34,35]. Common targets of immunoaffinity-based capture are tetraspanins (especially CD9, CD63 and CD81), accepted by MISEV2018 as a marker of EVs, as well as new markers such as EPCAM and EGFR, also used for capturing specific EV subtypes. The main limitation of this approach is the lack of availability of an antibody with high avidity for the target or antigen of interest, although this method is a valid and faster alternative for the isolation of EVs without the use of expensive instruments such as for differential centrifugation [34]. It is also important to consider that no antibody panel is able to exclusively recognize a selected subpopulation of EVs, as well as universal markers for one or more EV subtypes that have not yet been identified [8,18]. Many commercial kits, such as ExoQuick (System Biosciences LLC, Palo Alto, CA, USA) and Total Exosome Isolation reagents (Thermo Fisher Scientific Inc., Waltham, MA, USA) use PEG for EV isolation. Precipitation with PEG makes it possible to simultaneously process a large number of samples which makes this method attractive for clinical settings. The main disadvantage of this approach is the variable contamination of the precipitate with non-EV material [13].

As all the methods described above have limitations that do not allow the complete isolation of the EVs or complete purification of the EVs subpopulations without contamination, ISEV recommends using multiple separation methods in combination.

## 4. Molecular Mechanisms Involved in EVs Biogenesis

### 4.1. Exosomes Biogenesis

As mentioned above, the mechanism of biogenesis is one of the canonical characteristics used to define the three main classes of EVs [37,38]. The initial step of exosome biogenesis is the formation of early endosomes by the inward budding of the plasma membrane or, in some cases, from the trans-Golgi network (TGN), endoplasmic reticulum (ER) and mitochondria, contributing to the content of the early endosome which includes proteins, small molecules (including lipids and metabolites) and monoatomic ions [39,40]. The maturation into late endosome goes ahead with the invagination of the endosomal membrane and consequently, with the formation of intraluminal vesicles (ILVs), that are the future exosomes. The MVBs can undergo two different fates: they can fuse with the plasma membrane and release the exosomes into the extracellular space or alternatively, they can fuse with lysosomes/autophagosomes for degradation, in order to recycle their contents to be used as fuel for cellular metabolism [41]. The mechanisms that determine the fate of MVBs are still unknown (see chapter below). Two different mechanisms have been found to regulate the biogenesis of MVBs: the endosome sorting complex required for transport (ESCRT)-dependent pathways, which includes canonical and non-canonical pathways; and ESCRT-independent pathways.

The ESCRT machinery comprises four distinct complexes (Figure 1a(i)): ESCRT-0, ESCRT-I, ESCRT-II, ESCRT-III sub-complexes and ATPase complex (VPS4A-B, LIP5), as well as several accessory proteins (ALIX, HD-PTP) (for the mechanism, see Figure 1a). The knowledge of the ESCRT machinery of ILV biogenesis opens an avenue to understand and modulate the formation of exosomes through the manipulation of the ESCRT components.

Other parallel ways, termed non-canonical ESCRT-dependent pathways, can also achieve MVB production: HD-PTP protein binds ESCRT-0, -I and -III, bypassing the need for ESCRT-II [42]; syndecan-syntenin-ALIX pathway is independent of ubiquitination and ESCRT-0, but dependent on ESCRT-III and VSP4 required for the scission step (Figure 1a(ii)) [43]. Interestingly, in this pathway, heparanase represents an important extracellular modulator of exosomes production interacting with syntetin-1 and ALIX protein. Syndecan and heparan sulfate cargo are internalized and during endosome maturation, heparanase trims heparan sulfate on syndecan. This cleavage accelerates the exosomal secretion of syntetin-1, syndecan and other exosomal cargo [44].

Recent evidence has suggested that the ESCRT-independent processes have a role in the MVBs production [45]. These mechanisms are based on the presence of lipid rafts inside the endosomal membrane (Figure 1a(iii)): for example, the high levels of ceramide and other lipids in these domains drive the lateral segregation of cargo into specialized exosomes [46]. An additional ESCRT-independent mechanism of ILV formation is provided by proteins of the tetraspanin family (CD9, CD63, CD81, CD82, and CD151) that have the ability to organize membrane domains (termed tetraspanins-enriched microdomains, TEM), to regulate membrane shape and actin polymerization [47,48].

Interestingly, it has been observed that exosome secretion might be tightly regulated by the posttranslational modifications of proteins and lipids in the glycocalyx. In fact, in a mouse melanoma B16 model, the inhibition of N-glycosylation by metabolic manipulation is able to inhibit the biogenesis of non-exosomal vesicles (i.e., vesicles with distinct cargo molecules compared to exosomes) [49]. Moreover, an important aspect to be noted is that not only the glycocalyx composition but also the entropic forces generated by itself are able to regulate vesicle secretion generating projection extensions from the cell surface [50].

Which of these mechanisms becomes involved in exosomes biogenesis depends on the cargoes and on the cell type, and can be influenced by other signals and pathological stimuli that the cell can receive.

### 4.2. Microvesicles Biogenesis

The mechanisms underlying MVs formation act on the lipid composition and organization of the cytoskeleton to regulate membrane fluidity and deformability (Figure 1b). In contrast to the initial steps of exosomes biogenesis, MVs directly arise from the outward budding of the plasma membrane and are released into the extracellular space. One mechanism of their synthesis involves the ESCRT machinery, sharing part of the same molecular mechanism with exosomes [51]. Moreover, ESCRT components have been found in the plasma membrane of MVs, but how these molecules promote their formation is still unclear [52].

Calcium signaling, ATP, and lipid-mediated mechanisms can also induce the production and release of MVs. Several calcium-sensitive proteins can regulate MVs biogenesis: specifically, gelsolin and calpain can synergistically act to remodel cytoskeleton and membrane blebbing [53]. Furthermore, Annexin-2, a protein involved in calcium-induced exocytosis, plays a role in MVs formation [54]. Calcium also regulates, in an opposite way, the aminophospholipid translocase and lipid scramblase, enzymes involved in regulating plasma membrane asymmetry [55]. The release of ATP in the extracellular space induces the activation of purinergic receptors and of the acid sphingomyelinase, aSmase. This enzyme translocates to the plasma membrane outer leaflet, where it increases the efflux of cholesterol and membrane fluidity, thus facilitating membrane shedding [56].

Interestingly, increasing evidence suggests that MVs production is tightly associated with metabolic changes. For example, Lima and colleagues compared a normal melanocyte-derived cell line with a melanoma counterpart, revealing a two-fold increase in MV shedding by the malignant cells [57]. Moreover, in highly aggressive cancer cells, non-apoptotic plasma membrane blebs are frequently observed [58].

### 4.3. Mechanisms Involved in EV Secretion: From Cargo Sorting to EV Release

While the release of MVs is directly subsequent to their generation and fission, the release of exosomes requires additional steps that include the cargo sorting to MVBs and then to ILVs, the targeting to the plasma membrane and the exosomes secretion.

How cytosolic substances are incorporated into MVBs remains largely unknown. It may involve chaperon proteins, such as Hsc70 [62], or cargoes from the Golgi apparatus or from the plasma membrane during endosome maturation [63]. Other proteins which potentially play a role in protein sorting to exosomes are identified using quantitative mass spectrometry and these include Hsp90, 14-3-3 epsilon, and PKM2 [64]. Moreover, the mechanism that controls RNA sorting into MVBs are only beginning to be understood. Recent observations suggest that RNAs in EVs share the same motif sequences that may be targeted at EVs via RBPs (RNA-binding proteins) [65]. The finding that ESCRT-II is an RNA binding complex opens the possibility that it may also function to select RNA for incorporation into EVs [66].

Once formed, MVBs may progress through one of two maturation steps: they can fuse with the plasma membrane for exosomes release into the external space, or can reach lysosomes/autophagosome for their content degradation [39]. Even if the mechanisms that determine the fate of MVBs are still largely unknown, in recent years it has been observed that the balance between these two processes in cancer cells has shifted towards exosomal cargo release [67].

The endosomal sorting machineries of MVBs and their post translational modifications seem to be the first level of regulation for this balance; for example, the ISGylation of the TGS101 and of the tetraspanin 6 negatively regulates exosome release by promoting the fusion of MVBs with lysosomes [68]. Even if the release of exosomes may be constitutive, the balance can also be influenced by alterations in the physiological state of the cell, such as in lysosome storage diseases [69]. A similar balance exists between exosome secretion and autophagosome, and it depends on external stimuli, such as starvation [70].

Whatever is the MVBs’ fate, the transport and the fusion are two required steps; however, the effectors involved in these processes are certainly distinct. In general, intracellular transport involves the interaction with the cytoskeleton (actin and microtubules), molecular motors (dynein, kinesins and myosin), and small GTPases. For example, Rab7, a small GTPase, is involved in targeting MVBs to lysosomes by the recruitment of the retrograde molecular motor dynein, but interestingly, it is also required for exosome release [43]. The ubiquitylation status of Rab7 seems to be responsible of these dual exosomes’ fate [71]. There is significant evidence that the lipid and protein membrane composition of MVBs are essential for exosomes’ fate: indeed, the recruitment of Rab7 for lysosomal degradation is stimulated by cholesterol-limiting at the membrane, whereas the ILVs enriched in cholesterol preferentially reach out to exosome secretion [72]. Rab27 isoforms are also essential for exosomes secretion: the silencing of Rab27A induces a decrease in EV secretion in tumor cell lines [73]. Interestingly, the regulation of exosome secretion by Rab proteins are cell-type-dependent [74].

The finale step of exosomes secretion requires the fusion of MVBs with the plasma membrane to release ILVs as exosomes. This process is driven by NSF attachment protein receptor (SNARE) proteins (SNAP23, VAMP3-7, YKT6, SYX-5) and synaptotagmin family members [75]. It should be noted that the diversity of regulators involved in exosome secretion may depend on the organism, the cell type or the MVBs subtype. The complex composed by SNARE protein VAMP7 on the lysosome, syntaxin 7 on the plasma membrane and the lysosome regulatory protein synaptotagmin 7, regulate exosome secretion in some cells, like the human leukemia cell line K562, but not in others (MDCK cells) [76,77]. Recent work also demonstrated that pyruvate kinase type M2 (PKM2) promotes exosome release via phosphorylating SNAP23 protein in tumor cells [78]. Moreover, additional SNARE proteins, syntaxin 1A in *C. elegans* and YKT6 in *Drosophila*, are required for the secretion of specific EVs that contain Evi and Wnt proteins, respectively [79]. Better understanding the mechanisms that regulate the docking and fusion of MVBs with the plasma membrane is important to develop new tools and methods to follow and manipulate these processes.

The release of MVs requires a mechanism that is dependent on the interaction of actin and myosin, leading to vesicles budding off from the membrane of cancer cells [80]. Interestingly, TSG101 and the ATPase VPS4, involved in exosome generation, are also reported to participate in the formation and release of MVs [51]. Several reports support the involvement of cell signaling pathways in MV release: increased Ca^2+^ concentration induces MVs release by activating scramblase and calpain.

## 5. Role of EVs in Cell–Cell Communication

### 5.1. Mechanisms of Uptake of EVs by Recipient Cells

Depending on the cell type, EVs can exert their effects on recipient cells in two ways: they can initiate intracellular signaling pathways remaining bound to the cellular surface via ligand–receptor interaction, or they can be internalized to directly transfer proteins, nucleic acids and lipids to target cells (Figure 1c) [59,81]. The first mechanism is determined by specific interactions between proteins enriched at the surface of EVs and receptors at the plasma membrane (Figure 1c(i)). These interactions depend on the type of recipient cell: for example, in the antigen presentation, the EVs derived from B lymphocytes and dendritic cells present the major histocompatibility complex (MHC) to T-cells inducing immune responses by activating the T-cell receptor [26]. Several mediators of these interactions are known, such as tetraspanins, integrins, lipids, lectins, heparan sulfate proteoglycans and extracellular matrix (ECM) components, but the underlying molecular mechanism that regulates the specific targeting to recipient cells is still unclear [82].

The second way of EV uptake includes the cellular internalization pathways and the membrane fusion, which both lead to the transfer of EV content into recipient cells (Figure 1c(ii–iv)). The mechanisms by which EVs are internalized have been largely investigated: they can transfer their cargoes by clathrin-dependent or clathrin-independent endocytosis, such as macropinocytosis and phagocytosis, as well as through endocytosis via caveolae and lipid rafts [83]. While the fusion of EVs with plasma membrane directly leads to the release of their contents into the cytoplasm of recipient cells, endocytosed EVs can reach MVBs via the canonical endosomal pathway, and in most cases, they can be degraded after the fusion with lysosomes [84]. Alternatively, the internalized vesicles can escape the lysosomal pathway by back fusion with the MVB membrane releasing their cargoes into the cytoplasm of the recipient cell, a process that is poorly understood but important for delivering their contents, such as microRNAs, mRNAs, long noncoding RNA, genomic and mitochondrial DNA, but also proteins and lipids [85]. Several studies demonstrated that EVs are equipped with a unique tropism in the receptor-mediated endocytosis pathway. For example, it has been observed that tetraspanins CD9 and CD81 in EVs membrane interact with cellular integrin α_V_/β_3_ contributing to EV uptake into dendritic cells [86]. Furthermore, tetraspanin 8 and CD49 on EV membrane compose a complex that allows EV internalization into endothelial cells by the ligand CD54 expressed on recipient cells [87]. This observation opens the possibility to control the tropism of EVs for target cells. However, the heterogeneity of EVs and of coexisting uptake paths makes everything as difficult as possible.

### 5.2. Effects of Stromal-Derived EVs on Cancer Cells

Unlike what has been believed to date, Luga and colleagues demonstrated in 2012 that fibroblast-secreted exosomes induce the activation of Wnt-planar cell polarity signaling, promoting the invasive behavior of breast cancer cells [88]. This evidence goes against what was thought about tumor–stroma communication in cancer, namely a unidirectional process where tumor-derived EVs alter the behavior of stromal cells. Since then, several studies have shed new light on the important role of stromal-derived EVs in multiple aspects of cancer progression, tumor growth, invasion and metastasis, therapy resistance and tumor evasion of immunosurveillance. Stromal cells include fibroblasts, endothelial cells, bone-marrow mesenchymal stromal cells (MSCs) and immune cells. For example, in recent years, the role of MSCs in the tumor microenvironment has been increasingly studied. In this contest, an impressive study revealed that whereas multiple myeloma (MM)-MSC-derived EVs promote tumor growth, EVs from normal bone marrow-derived MSCs inhibit cancer cell proliferation, making them an attractive option for cancer therapy [89]. Moreover, it has been observed that stromal communication with cancer cells can influence treatment response: EV-associated RNA molecules induce resistance to chemo- and radiation- therapy in breast cancer cells by stimulating the pattern recognition receptor RIG-I to activate the STAT1 and NOTCH3 pathways [90]. Moreover, the EV-associate miR-19a molecule from astrocytes is involved in PTEN downregulation leading to metastatic outgrowth when human breast cancer cells disseminate to the brain, but not to another metastatic site [91]. Finally, recent work reported an emerging role of mitochondrial DNA in EV communication involved in the development of drug resistance [92].

### 5.3. Effects of Tumor-Derived EVs on Normal Cells in the Tumor Microenvironment

As mentioned previously, tumor-derived EVs (tEVs) exert an important effect on local or recruited stromal cells. Several studies have shown that tEVs can affect the behavior of normal fibroblast, endothelial cells, adipocytes, MSCs and immune cells. A critical stromal component susceptible to the effect of tEVs is represented by fibroblasts. In various cancer contexts, tEVs can induce the cancer-associated fibroblast (CAF) phonotype. For instance, prostate cancer EVs can induce proangiogenic and invasive CAFs from bone marrow MSCs [93]; and bladder cancer EVs can induce CAFs by promoting epithelial–mesenchymal transition (EMT), increasing the expression of mesenchymal markers and enhancing the migration and invasion of the recipient urothelial cells [94]. Ten years ago, it was thought that TGFβ was the molecule responsible for the differentiation of normal fibroblasts into CAFs [95]. Interestingly, a recent study demonstrated the existence of a bi-directional crosstalk between fibroblast and CAFs and how this communication was beneficial to tumor metastasis: if the tEV-associated TGFβ induces fibroblast differentiation into CAFs, likewise, the EVs derived from CAFs promote EMT, cell growth, migration and invasion in the bladder cancer cells by the secretion of IL-6 [96]. Several studies have also placed much emphasis on the potential role of microRNAs transferred by EVs in the switch of normal fibroblasts to CAFs [97,98]. For example, breast cancer patients have expression levels of miR-21, miR-378e and miR-143 in CAF-derived exosomes higher than normal fibroblasts-derived exosomes. This characteristic could promote the stemness and EMT of these cells [99]. In addition, studies demonstrated a role of CAF-derived exosomes in promoting drug resistance [100].

tEVs can also have a significant impact on the function and the behavior of endothelial cells: it was demonstrated that the incorporation of EGFR-containing EVs by endothelial cells leads to the activation of the MAPK and Akt pathway and triggers the endogenous expression of vascular endothelial growth factor (VEGF) signaling, stimulating tumor angiogenesis [101]. In addition, tEVs can induce the mimicking of the status of the cancer cells from which they derive; an interesting study revealed that tEVs released under hypoxia are enriched in hypoxia-regulated RNAs (miR-23a) and proteins that can be transferred not only to endothelial cells within the hypoxic region, but also to endothelial cells located in distant normoxic regions. In the normoxic endothelial cells, exosomal miR-23a induces the accumulation of hypoxia-inducible factor-1 α (HIF-1 α) and inhibits tight junction protein ZO-1, leading to enhanced vascular permeability and cell migration [102].

Extensive evidence suggests that tEVs mainly mediate an immunosuppression activity during tumor progression. Several mechanisms by which tEVs mediate immunosuppression have been proposed. EVs expressing Hsp72 activate immature myeloid cells with the ability to suppress T-cell activation (MDSC) by inducing the IL6/Stat3 signaling pathway in a Toll-like receptor 2 (TLR2)-dependent manner [103]. Moreover, tEVs containing death ligands, such as Fas ligand or TNF-α, directly induce cell death in immune cells binding the death receptor family members TNF receptor 1 (TNFR1) and the Fas receptor (FasR), [104]. Transforming growth factor-β (TGF-β) is one of the immunosuppressive targets found on the surface of tEVs: in patients suffering from acute myeloid leukemia and breast cancer; it is involved in the suppression of natural killer (NK) cell function and T-cell proliferation [105]. In vitro studies demonstrated an involvement of nucleic acids exchanged via EVs in the suppression of an immune system. For example, in pancreatic cancer, it has been observed that MHC II transcription factor regulatory factor X-associated protein (RFXAP) was inhibited by miR-212-3 transferred from pancreatic cancer-secreted exosomes [106]. Muller and colleagues discovered that regulatory T-cells exposed to tEVs in vitro show a downregulation of genes involved in the adenosine pathway, which induces a high expression of CD39 and enhanced adenosine production [107]. The high levels of nucleoside adenosine attenuate local immune responses [108].

Notably, the immunosuppressive potential is not the only effect of tEVs described: tEVs can also exert an immunostimulant role, carrying costimulatory molecules such as MHC class I and class II, cytokines and tumor-associated antigens (TAAs). It is not clear how tEV can induce dual response in immune cells: it depends on multiple aspects, such as the type of recipient cells, conditions in the tumor microenvironment (TME), and the cargo of inhibitory or stimulatory signals. Based on recent findings, it is interesting to note that the administration of anticancer vaccines together with tEV induces immunostimulatory effects [109]. For instance, costimulatory receptors CD80 and CD86 and MHC II molecules are upregulated in mice treated with dendritic cells loaded with EVs derived from glioma cells more than those treated with dendritic cells loaded with tumor cell lysates [110].

The immunostimulatory effect of tEVs in the TME is mediated by M1 macrophages, increasing the release of cytokines, such as IL-6, IFN-γ, TNF-α and IL-12, and promoting the T-cell-mediated immune response [111].

### 5.4. Effects of Tumor EVs on the Propagation of Tumor Heterogeneity

The intra-tumor heterogeneity is one of the major characteristics of neoplastic lesions. In this context, the EVs are involved in the transfer of cancer-derived signals between different tumor cell subpopulations and this function has important consequences. For example, Al-Nedawi and colleagues have demonstrated that tumor cells within glioma transfer each to other the oncogenic EGFRvIII receptor via tEVs [112]. This event leads to the activation of EGFRvIII downstream signaling pathways, such as MAPK and Akt cascades, in recipient tumor cells. The release of EVs from aggressive breast cancer cells, and subsequently, the metastatic potential of less-malignant tumor cells due to functional RNA transfer by tEV have been directly visualized. Moreover, this tumor-to-tumor communication is particularly relevant in the presence of specific microenvironmental conditions, such as hypoxia. Indeed, a hypoxic microenvironment stimulates tumor cells to release miR-21-rich exosomes that can promote the invasion and metastasis of recipient normoxic tumor cells [113]. Finally, recent studies performed genetic and proteomic analysis in phenotypically diverse subpopulations of patient-derived glioblastoma stem cells and found cell populations with intermediate phenotypes. This indicates that tEVs not only transfer malignant traits between cancer cell subpopulations, but intra-tumoral exchanges of microRNA/protein increase the heterogeneity of glioblastoma stem cells [114,115].

### 5.5. The Role of EVs in the Oncogenic Process: From Tumor Development to Cancer Resistance

As mediators of communication not only between tumor cells themselves, but also with surrounding stromal and immune cells, tEVs modulate a variety of processes involved in cancer progression.

To date, several studies have reported an involvement of EVs in the early stages of tumor development. Indeed, the activation of MAPK/ERK, PI3K/AKT and Wnt pathways mediated by EVs was verified in several cancer cell types. These two pathways are critical in mediating survival signals in a wide range of cell types. For example, in melanoma, the transfer of receptor PDGFRβ by EVs from melanoma donor cells leads to an activation of the PI3K/AKT pathway on BRAF-mutated recipient cells. This contributes to cellular proliferation and the inhibition of apoptosis [116]. Moreover, bladder cancer cell-derived exosomes increase the expression of phosphorylated Akt and ERK1/2, promoting cell proliferation and inhibiting apoptosis in recipient cancer cells [117]. Recently, Scavo and colleagues demonstrated that EVs transfer Wnt receptor Frizzled 10 to sustain the cell proliferation of colorectal, gastric, hepatic and bile duct cancer cells [118]. tEVs also carry many oncoproteins and oncomiRs internalized by target recipient cancer cells: for instance, miR-93-5p transferred by esophageal cancer cell-derived EVs increases cell proliferation of recipient cancer cells [119]. Moreover, miR-1246 and miR-205 were found to be overexpressed in EVs released by breast cancer cells and by cholangiocarcinoma cells, respectively [120,121].

EVs are also emerging keys players in the development of de novo vasculature, essential for tumor growth and metastasis. The upregulation of VEGF on the recipient endothelial cell induces the angiogenesis process and it was caused by the presence of promoting-angiogenic factors on the cargo of EVs, such as lncRNA CCAT2, lncRNA POU3F3, miR-21 or CXCR4 receptor [122,123,124]. In the cargo of EVs, it has also been found that VEGF, which contributes to angiogenesis stimulation [123]. Multiple myeloma-derived EVs carry a class of small non-coding RNAs, Piwi-interacting RNAs (piRNAs), which is delivered to endothelial cells to promote the proliferation, angiogenesis and invasion, and enhances their secretion of VEGF, IL-6, ICAM-1 and CXCR4, all of which are markers of the malignant transformation of endothelial cells [125]. Interestingly, EVs released by lung cancer cells during radiation therapy increase levels of miR-23a, which mediates a decrease in tumor suppressor PTEN; it is well known that PTEN takes part in the PI3K/Akt pathway, which prevents cell proliferation and suppresses vascular formation. The reduction in PTEN by miR-23a promotes endothelial cell angiogenesis, suggesting that this could be a resistance mechanism to this type of therapy [126].

Several studies underline the involvement of EVs in the stimulation of cancer cell migration and invasion. For example, in colon cancer, Wnt5b-associated exosomes promote cancer cell migration and proliferation in a paracrine manner [127]; hepatocellular carcinoma (HCC)-derived exosomes deliver SMAD3 protein and mRNA to detach HCC cells and facilitate their adhesion [128]. Moreover, colon cancer cell EVs were also shown to induce hepatocellular cancer cells migration via the activation of the MAPK/ERK pathway in recipient cells [129].

The role of tumor-derived EVs on reprogramming cancer cell metabolism has also been shown. For example, EVs derived from leukemia or lung cancer cells induce a decrease in the pentose phosphate pathway and an increase in glycolysis, leading to the multidrug resistant phenotype [130]. Glutathione S-transferase P1 (GSTP1), an enzyme capable of detoxifying damaging chemicals from cells [131], is contained at higher levels into EVs from doxorubicin-resistant breast cancer cells than EVs from patients who responded to the treatment [132]. Moreover, cancer-cell-secreted miR-122 facilitates metastasis by suppressing glucose uptake into non-tumor cells and thus increasing nutrient availability in the premetastatic niche [133].

We can conclude that tumorigenic changes in cancer behavior can also occur following the deregulation of signaling pathways by EVs. Several pathways seem to be involved: the Wnt signaling pathway in colorectal cancer or breast cancer; the TGF-β signaling pathway, in gastric or endometrial cancer; and the EGFR/human epidermal growth factor receptor (Her) signaling pathway in epithelial tumors. In addition, VEGF signaling controls angiogenesis, and its dysregulation has been implicated in metastatic colorectal cancer, renal cell carcinoma and non-small-cell lung cancer.

## 6. EVs Use as Liquid Biopsy

The use of EVs as liquid biopsy is a fast-growing field of research. In solid tumors, liquid biopsies include three major categories of biomarkers based on the circulating source of tumor-derived materials in biofluids, namely (1) circulating tumor DNA (ctDNA); (2) circulating tumor cells (CTCs); and (3) tumor-derived extracellular vesicles (tEVs). Plasma and serum are the most common body fluids used for liquid biopsies in solid tumors as they are quite easy to collect.

EVs have some advantages over the other two liquid biopsy biomarkers, since they: (a) are produced and released in huge amounts compared to ctDNA and CTCs; and (b) are stable and their lipid membranes protect their molecular cargoes [3]. To date, the most widely used EVs in liquid biopsy are the small EVs (exosomes), however, other EV populations have also recently been gaining interest, such as large oncosomes.

Since 2015, EVs have been used to distinguish between healthy subjects and patients with benign or malignant tumors. In fact, Melo and colleagues demonstrated that GPC1-positive EVs detected in sera could distinguish between pancreatic ductal adenocarcinoma (PDAC) patients, benign pancreas disease and healthy subjects [134].

EVs could allow obtaining information about tumor differentiation and/or to predict overall survival. Badovinac and colleagues demonstrated that, in the preoperative plasma of PDAC patients, EV size is associated with tumor differentiation. In fact, larger EVs are associated with metastatic PDAC, however, a difference of about 20 nm that could discriminate between patients with poorly differentiated and well/moderated differentiated tumors has also been demonstrated. Moreover, they have shown how a change in EV concentration and size was associated with overall survival [135]. Recently, Tao and colleagues compared the lipid composition of serum EVs from 20 pancreatic cancer patients and healthy controls [136]. This study highlighted that PE 16:0/18:1 was associated with the tumor stage, and this lipid was also found to be significantly correlated with patient overall survival.

In prostate cancer, Park and colleagues used PSMA expression for the enrichment of tEV from patients with benign prostatic hyperplasia or localized prostate cancer. They highlighted that the concentration of PSMA-positive EVs increased from low to high risk in prostate cancer patients [137]. Moreover, in exosomes, CK-8 expression and the presence of DNA methylation status were significantly correlated with a lower overall survival [138].

EVs surface proteins including CD63, PTK7, EpCAM, LZH8, HER2, PSA and CA25 could be used to distinguish prostate cancer from benign prostatic hyperplasia, as well as recurrence from no recurrence [139].

Large EVs (LEVs) are often found in metastatic prostate cancer [140]. This EVs population enables the increase in the ability to move and be invasive [141]. This population of EVs carries DNA that reflects genetic aberrations of origin cells, such as copy number variations [142]. Levels of LEV in the blood are associated with tumor load and bone marrow involvement in AVPC. The association of LEV levels with the CTC count improves the assessment of disease state [143]. In another recent study, the lipid profile of the urinary EVs of patients with stage 2 and 3 prostate cancer showed five metabolites (three ceramides, one glycerophospholipid and one acyl carnitine) that were significantly different between the two stages. In addition, there are non-significant differences in other metabolite families [144].

The EVs’ ability to cross the blood–brain barrier (BBB) and circulate in the blood constitutes a non-invasive method to monitor brain cancer treatment. In glioma, Osti and co-authors evaluated the clinical significance of plasma EV levels and showed that the concentration of EVs was higher in glioma patients compared to healthy subjects and decreased after surgery [145]. Sheybani and colleagues demonstrated that focused ultrasound hyperthermia increased the EVs released by glioma cells and changed their protein profile [146].

The proteomic study of plasma glioma-derived EVs was able to show dynamic tumor progression based on differential protein profiles [147]. In NSCLCs, the overall survival was correlated with NY-ESO-1, PLAP, EGFR, Alix and EpCam-positive EVs [148]. Other studies suggested that EVs allow, as a liquid biopsy biomarker, to track cancer evolution and drug resistance prediction.

In immunotherapy with anti-PD-1 monoclonal antibodies, the level of circulating PDL1-positive EVs before or during treatment could predict drug response. For this purpose, high levels of PD1-positive EVs before treatment draw attention to the fact that the T-cells cannot be reactivated using immune checkpoint inhibition. Instead, an early increase in PD1-positive EVs count during the treatment has a good correlation with T-cells reactivation and a good response to treatment. Notably, a lack of increase in PD-L1-positive EVs were associated with resistance to pembrolizumab [149,150].

In breast cancer, several studies highlighted EV involvement in drug resistance. In fact, in 2017, Ning and colleagues documented, in a cohort of patients treated with adjuvant anthracycline/taxane-based chemotherapy, that the presence of UCH-L1-positive EVs is associated with poor prognosis [151]. These phenomena are also shown with biological agent-based therapies (e.g., monoclonal antibodies). In 2012, Ciravolo and colleagues demonstrated that, in advanced breast cancer patients treated with trastuzumab, Her2+ EVs in serum behave as decoys; for this reason, the presence of these EVs could be used to predict patients’ response [152]. Moreover, EVs which transport lncRNA SNHG14 or TGF-β1 could be used for the same function; in fact, both populations of EVs are present in non-responding patients [153,154]. In prostate cancer, the presence of Arv7 mRNA in circulating EVs predicts resistance to hormone therapy and correlates with a short OS [155].

Ogata-Kawata and colleagues reported that the levels of seven different miRNAs in serum EVs were significantly elevated in colorectal cancer patients [156]. Furthermore, Matsumura and colleagues have shown that the presence of miR-19a-3p in serum EVs could predict the recurrence of colorectal cancer [157,158]. EV miRNAs have also been found to be promising biomarkers for the diagnosis of NSCLCs, because EVs have a large amount of tumor-derived RNA. Indeed, miR-660-5p, miR-17-5p and miR-126 are expressed at higher levels in NSCLC patients compared to healthy controls [159]. Moreover, miRNAs from circulating EVs seem to be associated with the stage, tumor grade, histology and prognosis of cancer patients; however, the sensitivity and specificity of these connections remained at only 60–80% [159]. In lung cancer, exosomes and their cargoes have also recently been studied with regard to drug resistance.

Another study demonstrated that the decreased expression of miR-29a-3p and miR-150-5p in small EV correlates with the delivered radiation therapy dose and might help to predict unexpected adverse events to radiation therapy. In advanced lung adenocarcinoma patients, plasma-derived exosome DNA has been successfully used in clinical genetic tests with TP53, EGFR, PKD1, and ALK [160].

The T790M mutation was found in EVs from patients treated with EGFR-TKIs which leads to gefitinib resistance [161,162]. The increased efflux of chemotherapeutic agents, which leads to decreased intracellular drug concentration, has been considered the main cause of drug resistance in cancer [163]. Tumor cells could also reduce the intracellular levels of cytotoxic agents by incorporating them into exosomes [164].

A new device, ExoSCOPE, measures the drug dynamics of EVs in patients’ blood samples. Recently, the authors highlighted that this device could distinguish distinct EV subpopulations with specific drug effects in cancer cells, and rapidly allow the identification of the treatment outcome [165].

To date, the use of EVs in liquid biopsy needs further study. It is necessary to complete analytical validation, to establish a reference range for healthy individuals, and a range for different disease conditions. After that, the next step will be to establish the clinical validity of EVs.

## 7. How Can EVs Be Exploited for Pharmacological Purposes?

The idea that EVs could not only be used as a potential biomarker, but also in the treatment of illnesses, is becoming even more consolidated. It is well known that chemotherapy remains the most common treatment for metastatic cancer, despite some limitations such as non-specific biodistribution, a lack of tumor cell-targeting, and low therapeutic index [166]. To overcome these limits, nanosized drug delivery systems are now developed. Nanotherapy, which involves the use of artificial nanoparticles loaded with low molecular weight compounds, therapeutic proteins or nucleic acids, includes several delivery devices, such as liposomes, micelles, dendrimers, polymeric or magnetic nanoparticles [167]. In this perspective, some features of the EVs are ideal as a drug carrier system for cancer treatment. EVs are not only able to easily penetrate cells membranes due to their small size, but also cross highly selective biological barriers such as the BBB [5,6]. In addition, EVs are associated with low toxicity and possess intrinsic characteristics of targeting and intercellular communication, giving them an important advantage as carrier systems [5]. Unlike artificial nanoparticles, the interaction between EVs and target cells depends, at least in part, on the specific ligands present on the surface of the EVs or embedded in their membrane [168].

### 7.1. EVs as Emerging Drug Carriers or Targets of Anticancer Therapy

The contribution of EVs to cancer pathogenesis has been studied for therapeutic purposes in two ways. The first way provides for their use as “Trojan horses”. In recent years, several studies have been conducted in this field and brought to light promising perspectives on cancer therapy.

Some researchers have evaluated the ability of EVs to deliver two broadly used conventional anticancer drugs, namely doxorubicin (DX) and paclitaxel (PTX). DX is one of the most effective anticancer drugs and is used alone or in combination regimens in the treatment of both hematological malignancies and various solid tumors. Unfortunately, the usefulness of DX-based therapy is highly limited by drug toxicity towards normal tissues, particularly cumulative cardiac toxicity [169]. Tian and coworkers recently evaluated the ability of engineered exosomes to target DX to tumor tissue [170]. These authors obtained tumor-targeting exosomes from mouse immature dendritic cells (DCs) which were engineered to express lysosome-associated membrane glycoprotein 2b (Lamp2b) fused with the tumor-targeting peptide iRGD (CRGDKGPDC) [171] and loaded them with DX by electroporation. The intravenous injection of drug-loaded engineered exosomes, but not drug-loaded untargeted exosomes or an equivalent dose of free DX, markedly reduced tumor growth, with no overt signs of host toxicity, in nude mice bearing MDA-MB-231 human breast cancer cell xenografts. Interestingly, mice receiving DX-loaded tumor-targeted exosomes had levels of serum markers of myocardial damage similar to those of untreated animals.

The use of the antimitotic agent PTX has become a broadly accepted therapeutic option in the treatment of various solid tumors. However, dose-dependent peripheral neuropathy and bone marrow suppression, as well as the development of drug resistance, limit the use of this agent. Moreover, to circumvent the poor hydrosolubility of PTX, its traditional injectable forms contain polyoxyethylated castor oil (Cremophor-EL), which would result in severe type I hypersensitivity reactions [172,173]. In this contest, Kim and collaborators developed an exosome-based formulation for the delivery of PTX [174]. Exosomes were collected from murine RAW 264.7 macrophages, loaded with the drug by different methods (i.e., simple incubation, electroporation, or sonication) and evaluated for their therapeutic potential. Interestingly, the loading of PTX into exosomes significantly increased drug cytotoxicity, as compared to free PTX, in both PTX-sensitive (wild-type Madin–Darby canine kidney; MDCK) and PTX-resistant (P glycoprotein-overexpressing MDCK) cultured cells. Further studies demonstrated that intranasal administration of PTX-loaded exosomes was more effective than a conventional formulation of free PTX (i.e., Taxol^®^) in mice bearing Lewis lung carcinoma (LLC) lung metastases. Interestingly, biodistribution studies demonstrated a near complete co-localization of intranasally delivered PTX-loaded exosomes with lung metastases. Recently, it has been demonstrated that the decoration of EVs with polyethylene glycol (PEG) results in stealth properties which significantly increase their circulation time in mice [175]. Based on these findings, Kim and coworkers developed a novel formulation of PTX-loaded exosomes incorporating aminoethylanisamide-PEG (AA-PEG), a molecular moiety capable of targeting the sigma receptor, a membrane-bound protein which is overexpressed by lung cancer cells [6]. Interestingly, the AA-PEG-vectorized exosomes loaded with PTX demonstrated a high ability to accumulate in LLC lung metastases upon intravenous administration and exerted a stronger therapeutic effect than non-vectorized PTX-loaded exosomes or free PTX (Taxol^®^). In another study, Pascucci and collaborators demonstrated that exosomes derived from mesenchymal stromal cells (MSC) primed with PTX have a strong anti-proliferative activity against CFPAC-1 human pancreatic cells. This work demonstrated that MSCs are able to package PTX, to deliver the encapsulated drug in the vesicles and to induce a reduction (50%) in tumor growth. They also proved that exosomes fused more easily with the plasma membrane of cancer cells compared with other nanoparticles [176]

In 2020, Yong-Jianf Li and colleagues used autologous exosomes derived from the human pancreatic tumor cell line Panc-1 as a system to deliver gemcitabine (GEM) to mice bearing xenografts of the same cell line. They demonstrated that GEM-treated EVs had the ability to escape the immune system, particularly phagocytosis, and to reach the target cancer cells. Interestingly, mice receiving exosome-delivered GEM neither developed metastatic lesions nor had signs of toxicity [177]. These findings are of considerable value in the light of the well-known drug resistance of pancreatic cancer and hepato- and nephrotoxicity of GEM.

Other studies demonstrated that exosomes loaded with chemotherapeutic drugs enable treating oral cancer. Rosenberger et al. investigated the therapeutic effect of menstrual mesenchymal stem cell (MenSC)-derived exosomes on hamster buccal pouch carcinoma and confirmed that the intra-tumoral injection of MenSC-exosomes leads to significant anti-tumor effects and tumor blood vessel loss. They found that the anti-angiogenic effects of MenSC-exosomes may have advantages in the treatment of oral squamous cell carcinoma (OSCC) [178].

A promising strategy against tumor progression results from engineered exosome-mediated siRNA that inhibit the post-operative metastasis of breast cancer (BC) [179]. The successful delivery of exosome-mediated antisense miRNA oligonucleotides against miR-21 improved the treatment efficacy for glioblastoma by inducing the expression of PTEN and PDCD4 and resulting in decreased tumor size [180]. Exosomes derived from androgen-sensitive human prostate adenocarcinoma cells carrying PTX negatively affected cancer cells’ viability [181]. Dendritic cell (DC)-derived exosomes in BC and macrophage-derived exosomes in lung cancer were loaded with the drugs, trastuzumab and paclitaxel, respectively, and successfully delivered to the recipient cells [6,182].

Exosomes have also been recently proposed by Sun and coworkers as a delivery system for the investigational drug curcumin [183]. Various preclinical studies indicate that curcumin, a phenolic compound isolated from *Curcuma longa*, has multiple biological activities, including antioxidant, antimicrobial, anticancer and anti-inflammatory effects. However, the clinical advance of curcumin has been hindered by its low water solubility, chemical instability, short biological half-life and low bioavailability after oral dosing [184]. Sun and colleagues demonstrated that encapsulation in exosomes increased water solubility, stability, as well as intraperitoneal and oral bioavailability of curcumin in mice. Moreover, they demonstrated that the intraperitoneal administration of curcumin encapsulated in exosomes was more effective than an equal amount of free drug in a murine model of lipopolysaccharide-induced septic shock [183]. More recently, Wu and co-workers showed that curcumin-loaded exosomes derived from H1299 lung cancer cells exhibit antitumor activity in vitro and are capable of upregulating transcription factor 21 (TCF21), a gene whose loss or reduced expression is a signature of malignant tumors in various cultured human lung cancer cell lines [185]. Collectively, these preclinical findings suggest that EVs display ideal features for anticancer drug delivery. However, further studies are needed to fully explore the potential of EVs as a delivery system for low-molecular weight cancer chemotherapeutics.

A second way to go in order to limit tumor growth and spread may consist of inhibiting EVs involved in drug resistance. In fact, recent studies highlighted as anticancer agents can be released from cancer cells in EV, resulting in an additional drug resistance mechanism. In this context, the EVs released from cancer cells and the possible influence of their biogenesis on the efficacy of cancer therapy are becoming ever more interesting.

In 2012, Marleau and colleagues removed HER2-positive breast cancer-derived EVs using a hemofiltration system. Since HER2-positive EVs promote chemoresistance and tumor growth, the clearance of this subtype of EVs prevents metastasis development [186]. Other researchers tried to remove EV using antibodies that recognize specific EV markers, for example CD9 and CD63, however, unfortunately, these depletion strategies also removed non-cancer-related EVs and could cause problems to healthy cells and to their functions [187]. Recently, another strategy in removing specific cancer EVs from the circulation was developed. This approach exploited, in lung cancer, aptamer-modified nanoparticles that are able to recognize EGFR-positive small EVs. The EV–nanoparticle complexes were eliminated through Kupffer cell uptake followed by bile secretion. In this work, the authors also observed cancer metastasis suppression in mice [188].

Finally, several agents are known to interfere with EV biogenesis and release and could therefore be used to reduce or reverse drug resistance and improve immunotherapy response. These compounds target different proteins and different stages of EV biogenesis. Among these, there are RAB27A inhibitors, nSMase inhibitors, and calcium channel blocking agents. Furthermore, several drugs which are clinically used to treat non-cancer diseases, including tipifarnib, ketoconazole, carbinol and simvastatin, were found to have the ability to inhibit exosome release. It is worth noting that some of these drugs affect exosome release by tumor cells but not by normal cells [189,190].

Recent studies demonstrated that EVs may mitigate the damage of chemotherapy or radiotherapy on normal tissues. For example, exosomes isolated from MSCs were used to improve tissue regeneration and wound healing. Montay-Gruel and colleagues demonstrated that the systemic delivery of human embryonic stem cell-derived EVs could improve the repair of lung tissue damaged by radiation therapy. In fact, EV-based therapy administered 24 h after radiotherapy interrupts the acute pathogenic cascade which was immediately activated post irradiation. This study also demonstrated a reduction in infiltrated immune system cells, such as macrophages, which are critical regulators of fibrosis, and highlighted the reduction in fibrosis and pulmonary density compared to control mice [191].

### 7.2. Clinical Trials

EVs have been extensively studied in many pre-clinical disease models [192], however, to the best of our knowledge, no EV-based therapeutic has been approved by a regulatory agency. To date, very few clinical trials have been conducted to evaluate EVs as therapeutic tools (see https://clinicaltrials.gov/), and most of these are phase I studies whose major goal was to obtain information about the safety of the approach.

The first clinical trial which evaluated EVs as a drug carrier started in 2005. This study tested the possible use of autologous DC-derived EVs pulsed with the melanoma antigen 3 (MAGE3) as a vaccination strategy in 15 patients with metastatic melanoma. No grade II toxicity was observed during the study. Based on these results, researchers at Gustave Roussy and Curie institutes developed an immunotherapy approach involving metronomic cyclophosphamide (mCTX) followed by vaccinations with tumor antigen-loaded dendritic cell-derived exosomes (Dex). mCTX inhibits Treg functions restoring T and NK cell effector functions and Dex are able to activate the innate and adaptive immunity. A phase I trial enrolled 41 unresectable NSCLC patients and showed the safety and feasibility of Dex vaccines, but no induction of T-cells could be monitored in patients. Since 2007, the same scientists developed and validated a new process for the isolation of second generation Dex with improved immune stimulatory capacities (NCT01159288).

In 2018, the NCT03608631 phase I trial started recruiting participants with the aim to find the optimal dose of mesenchymal stromal cells-derived exosomes containing KrasG12D siRNA (iExosomes). The study aimed to treat an estimated number of 28 patients with metastatic pancreatic cancer bearing the Kras G12D mutation. The researcher postulated that iExosomes might work better at treating pancreatic cancer.

Scientists at the James Graham Brown Cancer Center are going to test the benefits of plant-derived exosomes in head and neck cancer. In particular, trial NCT01668849 aims to explore the ability of grape exosomes, given to participants as grape powder, to act as an anti-inflammatory agent capable of reducing the incidence of oral mucositis triggered by the radio- and chemotherapy of head and neck tumors. In addition, the study evaluates the effects of grape exosomes on cytokine production, immune response to tumor exosomal antigens, as well as on metabolic and molecular markers. This study, which started in 2012, is still active, however, participants are not currently being recruited or enrolled, and preliminary results are not yet available. In 2011, the same institution organized a similar clinical trial (NCT01294072) to assess the possibility of using EVs as curcumin carriers in colon cancer patients; the results of this study have not been published.

Starting from the idea that EVs could be involved in the suppression of the immune system in cancer patients, an interesting clinical trial focusing on removing dangerous EVs was started. It is an early feasibility study (EFS), that investigates the use of the Hemopurifier^®^, to clear immunosuppressive exosomes, in combination with pembrolizumab (Keytruda^®^) in patients with advanced and/or metastatic squamous cell carcinoma of the head and neck (NCT04453046). Hemopurifier^®^ is designed to remove small EVs from the blood, operating as a blood filtration device similar to a kidney dialysis cartridge. The study will evaluate whether pretreatment with Hemopurifier^®^ before the administration of pembrolizumab is safe and well tolerated, and whether it leads to a lower number of exosomes in the blood (status: recruiting).

## 8. Conclusions

In this review, we highlighted the features and the possible applications of EVs in cancer management. Unfortunately, the isolation and identification of tumor cell-derived EVs still need standardizations and common guidelines [193].

Emerging data suggest that EVs could be a useful tool in clinical settings as diagnostic biomarkers, as well as for therapy. The theragnostic use of EVs could establish a completely new area of research in the field of cancer research and management. Therefore, the continuous in-depth investigation studies with large cohorts and clinical trials are re-quired. It is hoped that the continued development and refinement of EVs technology will further improve their potential clinical utility for cancer diagnosis, monitoring and treatment.

## Figures and Tables

**Figure 1 diagnostics-11-01118-f001:**
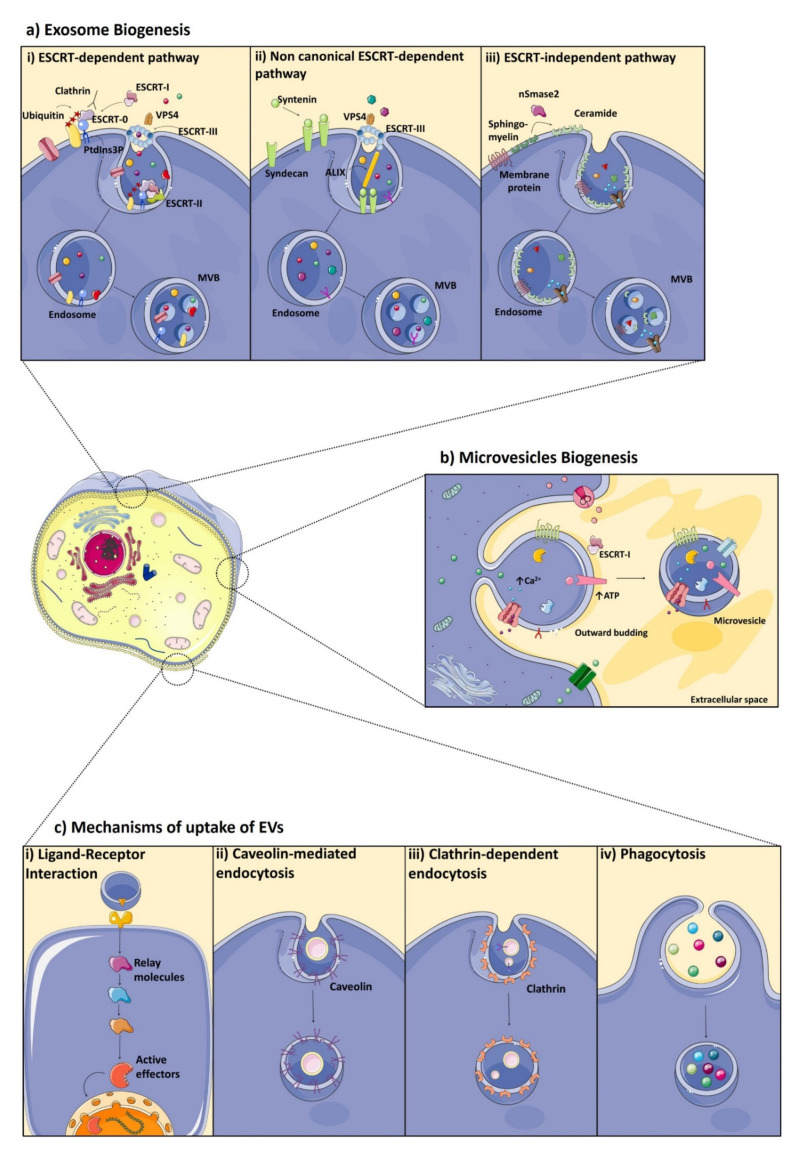
Extracellular vesicles formation: (**a**) exosome biogenesis. (**i**) the ESCRT-dependent pathway. In the canonical ESCRT-dependent pathway, phosphatidylinositol-3-phosphate (PtdIns3P) recruits the ESCRT-0 (STAM1, HRS) to the endosomal membrane by ubiquitinated proteins and is subsequently clustered into microdomains via clathrin binding. Then, the ESCRT-0 recruits ESCRT-I (TSG101, VPS28, VPS37A-D, MVB12A-B, UBAP1), via the direct interaction of HRS with the TGS101 subunit of ESCRT-I. Moreover, ESCRT-I also interacts with the ESCRT-II complex (EAP30, EAP20, EAP45); this interaction is thought to be responsible for membrane deformation into buds. The last complex that interacts with the ESCRT-II is the ESCRT-III (CHMP2-7, CHMP 1A-B), which is required for ILV scission into the MVB lumen and disassembled after ILV scission via AAA-ATPase Vps4 [59]; (**ii**) Non-canonical ESCRT-dependent pathways. Syndecan-syntenin-ALIX pathway. The transmembrane protein syndecan recruits syntenin, which interacts with ALIX. This protein recruits ESCRT-III for membrane budding and cargo sorting, and VPS4 for the scission step, which occur independently of ubiquitin and ESCRT-0; (**iii**) ESCRT-independent pathway. The neutral type II sphingomyelinase family (nSmase2) converts sphingolipids to ceramides, which can induce lipid curvature and trigger the conversion of ILVs into MVBs [60]. (**b**) Microvesicles Biogenesis. The mechanisms underlying MVs biogenesis are still being revealed: it has been observed that ESCRT molecules could promote their formation; calcium, ATP and lipid-mediated mechanisms can also induce the production and release of MVs. (**c**) Mechanisms of uptake of EVs. (**i**) Ligand–receptor interaction. In the immunomodulation, the EVs derived from cancer cells carry programmed death ligand-1 (PD-L1) at the surface, by which they bind the programmed cell death protein-1 (PD-1) and lead to cancer immune evasion [61]. Mechanisms that involve the cellular internalization pathways and the membrane fusion; (**ii**) Caveolin-mediated endocytosis; (**iii**) Clathrin-dependent endocytosis; (**iv**) Phagocytosis. Made using cliparts from Servier Medical Art by Servier, https://smart.servier.com.

**Table 1 diagnostics-11-01118-t001:** Characteristics of different types of extracellular vesicles.

	Exosomes	Microvesicles	Apoptotic Bodies
Size	30–120 nm	100–1000 nm	500–4000 nm
Biogenesis	Endolysosomal pathway; fusion of multivesicular bodies with the plasma membrane.	Shedding of plasma membrane with cellular content.	Plasma membrane budding of apoptotic cells.
Putative Markers enriched in the EVs	CD63, CD81, CD9, Hsp60, Hsp70, Hsp90, Alix, ESCRT components, Flotillin.	Integrins, selectins, CD40, CD9, CD63, CD81, CD82 and origin cell-specific markers.	Elevated phosphatidylserine.
Molecular Cargo	Proteins, RNA, miRNA and lipids.	Proteins, RNA, miRNA and lipids.	Organelles, proteins, DNA, RNA, lipids.

## Data Availability

Not applicable.

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
