# Peer review of "Cell-Secreted Vesicles: Novel Opportunities in Cancer Diagnosis, Monitoring and Treatment"

_diagnostics, 2021, doi:10.3390/diagnostics11061118_

Round 1

Reviewer 1 Report

This relative long review article describes many basic functions of extracellular vesicles and their role in cancers. The article is well written and interesting, but I think it could benefit from a bit of an extra editing. The main weakness is that the manuscript tries to cover so many different areas that some of them remain poorly or one-sidedly described. On the other hand, some part are very detailed. For example, the chapter 4 with detailed description of EV biogenesis seems to be a bit far from the main focus that is cancer, even though it is interesting basic-EV knowledge. 

2: EVs Classification and Characteristics chapter: This chapter does not have clear focus and it includes detailed descriptions of certain selected things. For example the anticoagulant used in plasma samples is described in details but no mention about lipoproteins or platelet EVs as contamination source for plasma EVs. The chapter could be cut a bit shorter and considered what are the relevant things and needed level of details regarding the main message of the review.

3. EVs: Separation methods chapter: Description of EV separation methods are too focused on ultracentrifuge-based methods: those are described in very detailed manner (could be cut a bit shorter) and some other relevant and commonly used methods are not mentioned at all. At least size-exclusion chromatography (SEC) and precipitation-based commercial kits should be mentioned as those are increasingly used methods. For example SEC was the second most used method in the survey in 2019. That is debatable whether mentioned filtration even is a separation methods as it mainly concentrates the EVs to smaller volume instead of purifying them. Also, it would be good to discuss about the suitability of the methods for clinical use as for example ultracentrifugation-based methods can not be used with larger sample numbers.

In 6. EVs use as Liquid Biopsy: The chapter concentrates mainly on the changes found in EV concentration and size. The reliable measurement of those is still very hard and even similar machine in different labs can give different results. There is still no clear knowledge about the variation of EV number/concentration even in healthy humans. The concentration/size can never be used as disease-specific biomarkers as several different conditions are known to change the EV profile. It is therefore debatable how relevant the size/concentration differences are regarding liquid biopsies. The most promising EV-related biomarkers would be more likely within the EV cargo, for example proteins, lipids and RNAs (mRNA/miRNA/other small RNAs). There is a large number of studies (and reviews) describing changes in those profiles in several different cancers and the chapter could outline those a bit more.

Line 66-67: The most used starting material was cell culture supernatant: add a note that this sentence applies only to biofluids if that was the point.

Line 124: Why refer to the survey in 2015 instead of 2019?

Acknowledgements and Conflicts of interest include text from the journal template.

Reviewer 2 Report

This review paper comprehensively describes the biogenesis, release, uptake, biological functions and clinical applications of EVs. 

Please find my minor comments on this manuscript.

Some of the key recent papers describing biochemical characterization of EVs are missing and need to be discussed in detail. These might include doi: 10.1016/j.cell.2020.07.009; doi: 10.1016/j.cell.2019.02.018; doi: 10.1016/j.cell.2019.02.029.

Although the intracellular machineries involved in the biogenesis of EVs are well-documented, extracellular factors, e.g., heparanase, O-glycans and N-glycans, that regulate this process are not discussed (doi: 10.1074/jbc.C112.444562; doi.org/10.1038/cr.2015.29; doi: 10.1016/j.cell.2019.04.017; doi: 10.1016/j.celrep.2020.108261). The authors might constructively include and discuss these findings in their revised manuscript.

Round 2

Reviewer 2 Report

This reviewer has no further comments on this manuscript.